# Speech errors in consecutive interpreting: Effects of language proficiency, working memory, and anxiety

**Nan Zhao**  [1]*, **Zhenguang G. Cai**[2], **Yanping Dong** [3]

1 Department of Translation, Interpreting and Intercultural Studies, Hong Kong Baptist University, Hong Kong, Hong Kong, 2 Department of Linguistics and Modern Languages/Brain and Mind Institute, The Chinese University of Hong Kong, Hong Kong, Hong Kong, 3 School of International Studies, Zhejiang University, Hangzhou, China

* nanzhao@hkbu.edu.hk

## Abstract

Interpreting can be seen as a form of language production, where interpreters extract conceptual information from the source language and express it in the target language. Hence, like language production, interpreting contains speech errors at various (e.g., conceptual, syntactic, lexical and phonological) levels. The current study delved into the impact of language proficiency, working memory, and anxiety on the occurrence of speech errors across these linguistic strata during consecutive interpreting from English (a second language) into Chinese (a first language) by student interpreters. We showed that speech errors in general decreased as a function of the interpreter's proficiency in the source (second) language and increased as a function of the interpreter's anxiety. Conceptual errors, which result from mistaken comprehension of the source language, decreased as a function of language proficiency and working memory. Lexical errors increased as a function of the interpreter's tendency of anxiety. Syntactic errors also decreased as a function of language proficiency and increased as a function of anxiety. Phonological errors were not sensitive to any of the three cognitive traits. We discussed implications for the cognitive processes underlying interpreting and for interpreting training.

## Introduction

Interpreting can be considered as a unique form of spoken language production [1], where the interpreter deciphers a message from the source and transmits it in the target language. Similar to spoken language production, interpreting is prone to a range of speech errors. Although extensive research has explored speech errors in standard language production, little is understood about the types of errors interpreters make when translating from the source to the target language. This paper investigates whether and how individual differences in interpreters' *language proficiency*, *working memory*, and *anxiety* affect the occurrence of speech errors during consecutive interpreting from English (second language, L2) to Chinese (first language, L1). This study provides pedagogical insights for interpreting instruction and training.

**Competing interests:** The authors have declared that no competing interests exist.

## Speech errors in language production

In speaking, the speaker transforms a pre-verbal message into a linguistic expression (e.g., a sentence; [2, 3]). This transformation involves several stages. Initially, the speaker conceptualises a pre-verbal message, the meaning intended to be conveyed to the audience. Following this, the speaker selects words that mirror the semantic components in the message, and a sentence structure to organize the words [4]. Ultimately, the syntactically arranged words must be phonologically encoded and articulated (e.g., [5]). It is evident that any of these stages can falter, resulting in errors at different linguistic levels [for reviews, see 6, 7].

A conceptual error transpires when the semantic information expressed does not correspond with the communicative intent (e.g., saying *good morning* at 13:00 in the afternoon). As per the cumulative semantic interference effect (e.g., [8]), speakers are slower and produce more conceptual errors in picture naming (e.g., using *tiger* to name a lion) when pictures belong to the same semantic category (e.g., pictures of animals) compared to different categories (e.g., pictures of animals, plants and furniture; see also [9]). Occasionally, a conceptual error arises because the utterance does not aptly serve a communicative function. For instance, a speaker might accidentally say *the star* to refer to a small star without realizing the presence of another larger star, or say *the small star* when there is only one star (e.g., [10, 11]).

A lexical error surfaces when an incorrect lexical expression is used. As an example, in a Stroop task where participants name the ink colour of a colour name (e.g., the word *red* printed in green), they sometimes erroneously say *red* instead of *green* due to the word *red* being highly activated (e.g., [12]). Hanley et al. [13] demonstrated that, in consecutive picture naming, children tend to use a related but incorrect name to describe an animal (e.g., *dog* or even *dat* to refer to a cat), and this tendency is stronger among younger children (e.g., children aged 5 versus children aged 7).

A syntactic error occurs when there is improper assignment of word order or grammatical functions, as in the case of *I left my briefcase in my cigar* (when *I left my cigar in my briefcase* is intended, [14]). In what is termed as broken agreement (e.g., [15]), native speakers of English can sometimes commit morphosyntactic errors in verb agreement (e.g., uttering *The key to the cabinets are rusty*). Ivanova et al. [16] also demonstrated that speakers can occasionally be primed to produce ungrammatical sentence structures such as *The dancer donates the soldier the apple*.

Lastly, phonological errors occur in spontaneous speech (e.g., [17]). Common phonological errors encompass anticipation (e.g., *leading list* instead of *reading list*), preservation (e.g., *a phonological fool* instead of *a phonological rule*), and phoneme exchange (*blake fruid* instead of *brake fluid*). In experimental settings, phonological errors can be readily induced by methods such as tongue twisters (e.g., [18]).

## Speech errors in interpreting

Interpreting, as a distinct form of language production, largely follows the same cognitive processes as regular language production [1]. In this, interpreters begin with a conceptual message, retrieve target language lexical expressions corresponding to the semantic elements in the message, determine a syntactic structure to organize these lexical expressions, and retrieve phonological information for the arranged and structured lexical expressions before delivering the target language speech. Therefore, it is anticipated that many speech errors found in regular language production (e.g., those previously reviewed) may also appear in interpreting.

However, interpreting notably deviates from regular language production in several crucial ways. First, interpreters commence with a conceptual message that they derive from the source language, rather than generating their own conceptual message. As such, interpreting might

be vulnerable to conceptual errors stemming from inaccurate comprehension of the source language, unlike regular language production.

Second, it has been established that bilinguals implicitly activate L1 translation equivalents when comprehending an L2 [19]. Further, evidence suggests that even in consecutive interpreting, interpreters prepare target language translation equivalents while listening to the source language [20–22]. This could lead to speech errors, where interpreters may prematurely settle on an inaccurate translation equivalent, particularly when a source language word is not used for its standard meaning (e.g., [23]). For instance, when interpreting *meter reading* into Chinese, the word *reading* may swiftly activate the Chinese translation *yuedu* (阅读) instead of the correct translation *dushu* (度数), resulting in a speech error. Indeed, there is evidence that student interpreters often inaccurately interpret by using a target language word that is an interlingual homograph of a source language word (e.g., *billion* between English and Polish; [24]).

Third, abundant evidence shows cross-language syntactic influence in bilingual language processing (e.g., [25, 26]), where a bilingual speaker's syntactic choices in one language are influenced by their recent syntactic experiences in the other language. Similar cross-language syntactic interference has been found in translation, where translators tend to replicate syntactic structures from the original text into the translated text (e.g., [27]). Given that interpreters often prepare target language syntax while comprehending the source language (e.g., [23, 24, 28, 29]), such cross-language syntactic influence might be commonplace. While this might facilitate language production in the target language when the two languages have similar structures, it could lead to syntactic speech errors when the syntactic sequence copied from the source language is illicit in the target language. For example, repeating the word order of a locative structure such as *He likes to study in the library* into Chinese would result in a syntactic error with the locative at the end of the Chinese sentence. An example would be, *ta xihuan xuexi zai tushuguan* (lit. he like study in library), instead a syntactically correct output where the locative precedes the verb, *ta xihuan zai tushuguan xuexi* (lit., he like in library study).

Lastly, there is evidence of a phenomenon known as cross-language phonological priming in bilingualism (e.g., [30, 31]), where phonological information in one language can activate shared or similar phonological representations in the other language. As a result, if phonological processing in the source language primes an inappropriate phonological representation in the target language, an interpreter might mispronounce words in the target language, leading to a phonological error.

## Interpreting and individual differences in cognitive functions

Interpreting differs from normal speaking because it is a complex cognitive task that requires simultaneous processing capacities under time constraints [32–34]. Interpreting requires the support of different cognitive functions. For instance, in consecutive interpreting, interpreters must first understand the source speech, retain the information in working memory (or sometimes in written notes), and then transcode the information into the target language. This is done under significant time pressure and often before a large audience. Successful interpreting thus demands cognitive supports such as proficient language skills, strong working memory, and the ability to manage anxiety under pressure and in front of an audience. All these cognitive functions have been extensively studied in interpreting research due to their relevance to interpreting performance (e.g., [34–38]).

Language proficiency, defined as a person's ability to use a language, especially an L2, is typically measured using standardized tests that assess various aspects of language knowledge and use, including reading, writing, speaking, and listening skills [37]. It is unquestionable that

language proficiency, especially in L2, is crucial for interpreting. Interpreters must sufficiently understand the source language to comprehend it and know the target language well enough to produce it. L2 proficiency determines how well the language is comprehended (e.g., [39, 40]), with more proficient L2 users being more fluent in their language production (e.g., [41]). Furthermore, proficiency in the second language also impacts first language processing because more proficient second language users are better at activating translation equivalents in the first language [42], a tendency that is beneficial in facilitating interpreting. Indeed, studies have shown that individual differences in language proficiency account for differences in interpreting performance [38].

Working memory is a cognitive function that temporarily holds and manipulates information for complex cognitive tasks such as language comprehension, learning, and reasoning [43]. It can be assessed in various ways; for instance, the reading span task requires participants to read sentences while remembering the last word in each sentence, and then recall all the final words in order at the end [44]. Working memory has long been assumed to underlie interpreting performance [see 45, for a review]. Zhao et al. [22] showed that interpreters make lexical predictions in source language comprehension, but not when their working memory resources are depleted. Christoffels, de Groot, and Kroll [46] showed that trained interpreters outperformed both bilingual students and English teachers in working memory but not necessarily language proficiency, thus suggesting that professional interpreting practice is particularly associated with better (verbal) working memory capacity. Relevant studies also provide evidence showing that higher cognitive capacity is associated with better performance on word translation tasks [47]. Moreover, a graphical models analysis indicated that working memory, together with language proficiency, had an independent effect on interpreting performance [35].

Another factor that impacts interpreting is anxiety, a feeling of unease, such as worry or fear, that people typically have toward stressful situations. Some individuals tend to be generally more anxious than others (a person's trait anxiety; [48]). As interpreting often requires public speaking, (trait) anxiety can become a major hurdle for interpreters, especially for interpreting students who lack experience in public speaking (e.g., [34]). Therefore, the capacity to control anxiety has traditionally been considered one of the requirements for interpreting [49–54] and a predictor of interpreting competence [36, 55]. Although empirical studies on the influence of anxiety on interpreting performance are scarce, there is a wide consensus that anxiety is inherent to interpreting–both in the consecutive and simultaneous mode–even though its impact is not clearly defined [56]. Interpreting research on anxiety has focused on the professional realm, concentrating mainly on physiological responses to stress during interpreting and on performance, including cardiovascular activity [51], causes of anxiety [49], and the relation between anxiety and quality in prolonged interpreting turns through chemical and physiological analysis [54].

Zhao [34] investigated the effects of language proficiency, working memory, and anxiety on interpreting performance and disfluencies (though she did not report on how these cognitive functions impact speech errors). She found that better L2 language proficiency, larger working memory spans, and lower anxiety levels all contribute to improved interpreting performance. More anxious interpreters were observed to produce more disfluencies in their target language output, including more fillers (such as *er* and *um*) and repetitions.

Individual differences in cognitive functions also seem to impact the occurrence of speech errors. A study by Shen and Liang [57] revealed that compared to student interpreters, professional interpreters are more conscious of semantic and syntactic errors and hence more likely to correct them. This suggests that the higher language proficiency in professional interpreters may assist in detecting speech errors. Working memory has also been shown to impact speech

errors in native language production, with speakers possessing lower working memory spans being more prone to producing speech errors [58, 59]. In addition, interpreters with higher working memory capacities were found to be more fluent in interpreting, with fewer repetitions and pauses [34, 60]. Anxiety levels also appear to impact the occurrence of speech errors, particularly in the form of stuttering. More anxious interpreters tend to be more disfluent in their speech, which can result in more speech errors [34, 61].

Despite these insights, it remains unclear how these cognitive functions might affect the occurrence of speech errors in interpreting. As previously mentioned, speech errors can occur at different stages of target language production, leading to conceptual, lexical, syntactic, and phonological errors. However, to our knowledge, no study has yet examined how these different stages of target language production in interpreting might be differentially affected by individual differences in language proficiency, working memory, and anxiety. The current study presents a first attempt to explore this issue. Specifically, it investigates how language proficiency, working memory, and anxiety might impact the occurrence of speech errors when student interpreters interpret from their first language (source language) into their second language (target language). We chose to focus on consecutive interpreting from L2 (English) to L1 (Chinese) by student interpreters for several reasons. First, target language production in consecutive interpreting, which occurs without concurrent comprehension, is more akin to spontaneous language production than simultaneous interpreting, where production and comprehension occur simultaneously. For instance, it is possible that interpreters in simultaneous interpreting make speech errors that are irrelevant to the cognitive functions examined here; for example, interpreters will often prematurely interpret a source language sentence due to time pressure, resulting in speech errors that are due to premature interpreting. Therefore, speech errors in consecutive interpreting may therefore provide clearer insights into the different stages of target language production and the influences of various cognitive functions. Second, using the first language as the target language in interpreting helps to eliminate effects on speech production that are a result of being an L2 speaker. Finally, we chose to focus on student interpreters, as the results of this research could have important pedagogical implications for interpreting training. The findings could guide interpreting trainers to focus on aspects of cognitive functions that tend to result in speech errors, such as emphasizing public speaking training to reduce anxiety.

## Method

This study presents new data from a project previously described in Zhao [34]. To provide a comprehensive understanding of the project, we include a detailed description below.

### Participants

The study was approved by the ethics committee of the Centre of Linguistics and Applied Linguistics at Guangdong University of Foreign Studies. A group of fifty-three senior students studying interpreting and translation at Guangdong University of Foreign Studies, 45 of whom were women, participated in a consecutive interpreting test during one of their course sessions. Each student was a native Mandarin Chinese speaker who had begun learning English as a second language during their primary education. They all pursued English as their major in college, showcasing proficient use of the language in both their academic pursuits and daily life. The participants had initiated their interpreting training in their third year of college, meaning they had accrued a year's worth of interpreting training prior to their participation in this study. Participants gave their informed written consent before participating in the experiment.

### English-Chinese consecutive interpreting test

Drawing from the methodology of prior studies on interpreting assessment [32, 50, 62], we employed an actual speech from an international conference on computer technology. This speech was adapted for interpreters to provide consecutive interpreting (see Appendix 1 for the adapted speech; https://osf.io/p64yf/). The original speech ran for approximately 10 minutes, delivering an average of 180 words per minute with a moderate information density. With an audio frequency ranging from 500 to 4000Hz, the speaker's pronunciation was clear, marked by a standard American accent, and the speech itself was delivered in a well-structured and logical manner. To facilitate consecutive interpreting, the speech was divided into segments, each c*omposed of* about 2–5 sentences. A break for interpreting followed each segment, which was 1.5 times longer than the segment preceding it, aligning with the standard practice for the China Aptitude Test for Translators and Interpreters for consecutive interpreting Level II).

The test took place in a multimedia classroom, the same setting where participants attended their interpreting classes. Each participant had access to a computer fitted with headphones that featured a microphone. As the test commenced, participants donned their headphones and were given blank sheets of paper for note-taking during the interpreting process. Throughout the test, they listened to the speech, broken down into manageable segments, and were permitted to jot down notes. Following each segment, an audible cue prompted them to begin their interpreting. The interpreting output was individually recorded, with the entire process lasting about 25 minutes.

### Transcription and error coding

The consecutive English-Chinese interpreting recordings of all 53 participants were transcribed, and any errors, amongst other details, were coded by the first author. These transcriptions and error codings were independently verified by another researcher conversant with the study's objectives (a sample transcription of the interpreting coding can be found at https://osf.io/p64yf/). As mentioned in the introduction, errors were categorized based on their linguistic nature—conceptual, lexical, syntactic, or phonological errors. Table 1 displays examples of each error type.

We accounted for the total number of errors and the tally of each error type for every participant. Given the varying lengths of the target speech (ranging from 1785–3318 characters, with an average of 2529 characters, and a standard deviation of 350), we calculated the number of errors (per type) per 1000 characters in the interpreting output. To exemplify, an interpreter with a conceptual error rate of 10 signifies that they committed 10 conceptual errors for every 1000 characters.

### Language proficiency test

All 53 participants additionally participated in a language proficiency test (in English, their L2) following the interpreting task. The test was structured on the foundation of the Test for English Majors Band 8, a nationally recognized English proficiency test designed for fourth-year English majors, mirroring the academic stage of our participants (for the test, refer to Appendix 2; https://osf.io/p64yf/). Considering that some test items did not directly assess language proficiency (e.g., items focused on linguistics and English literature), we excluded these sections. Instead, we incorporated segments evaluating reading comprehension, listening comprehension, and writing. These three components had a cumulative maximum score of 56. After the distribution of test papers and answer sheets, participants embarked on the

**Table 1. Classification of speech errors (target error in Chinese and its source in English are in bold).**

| Error type | Definition and example |
|---|---|
| Conceptual | Mis-comprehension of source language |
| | But what has happened is that these devices start getting fatter and fatter and fatter.<br>然而我们将, 然后我们今后的产品将会越来越 其特点将会越来越丰富.<br>(lit. But we will, then our in future products will be more and more, **their features will be more and more diversified**)<br>(Explanation: The output in bold does not match the source text in bold in meaning) |
| Lexical | Use of an incorrect translation equivalent word |
| | We want to make big displays, 0 defects, perfect color, very bright, large.<br>我们的显示器不允许出现任何闪失。我们的颜色会做得更加完美。显示也, 呃…….画面会更加明亮。<br>(lit. Our display will not allow for defects. The color will be more perfect. **Show** also . . . er . . . images will be brighter)<br>(Explanation: 显示 is another Chinese translation of the word display, but the word is not appropriate in the sentence here) |
| Syntactic | An error in syntax |
| | So everything in our corporate strategy is the opposite.<br>与……你提的要求与我们公司的战略正好相反。<br>(lit. **With** . . . your request is the opposite of our corporate strategy)<br>(Explanation: 与 is a proposition in Chinese, which was a syntactic false start here) |
| Phonological | Mispronunciation |
| | . . .let students check them out for *a science class* and put them back at the end of a class.<br>让他们在科学科, 科学课的时候用然后下课的时候归还……<br>(lit. Let them in scient, science time use and after class return)<br>(Explanation: 科 /ke1/is a preserved phonological error, the correct pronunciation should be /ke4/; numbers representing tones) |

listening comprehension section, followed by reading comprehension, and concluded with the writing component.

## Working memory test

In line with our established practice for gauging interpreters' working memory [63], we utilized a version of the reading span task, originally conceived by Daneman and Carpenter [44] and subsequently refined by Mizera [64]. During this test, participants memorized a list of words, subsequently crafting a sentence for each word (for word examples, see Appendix 3; https://osf.io/p64yf/). The test materials consisted of 100 two-character Chinese words, all of which were high in frequency as per the Modern Chinese Word Frequency Dictionary. The test items were grouped into five sets, each containing 2, 3, 4, 5, or 6 memory words per trial, respectively. Each set consisted of 5 trials, culminating in a total of 25 trials. During each trial, participants individually read the words displayed on a computer screen, with each word appearing for a single second. Following the display of words, the prompt "造句" ("make a sentence") appeared, at which point participants pressed the spacebar and verbally constructed a sentence for each presented word; they then pressed the spacebar again upon sentence completion. All verbal responses were digitally recorded. The order of trials was randomized. A practice session, comprising two trials—one with 2 memory words and another with 3—preceded the actual test. The working memory score was determined by the quantity of words successfully utilized in grammatically sound sentences.

## The anxiety questionnaire

Note that anxiety in interpreting can be a multifaceted element, encompassing a student interpreter's general day-to-day anxiety (e.g., when interacting with others or performing tasks) as

well as their interpreting-specific anxiety (e.g., subpar English proficiency or deficient memory). To omit factors related to language and memory (already measured by the language proficiency and working memory tests), we chose to employ a Chinese scale utilized by Dong et al. [65] (refer to Appendix 4 for the scale; https://osf.io/p64yf/). This scale is a fusion of methodologies from Zhang and Schwarzer [66] and Spielberger et al. [48]. The scale consists of two parts: Part 1 assesses self-efficacy anxiety (anxiety concerning one's competence for a task), while Part 2 examines state-trait anxiety (anxiety as a personal characteristic). Responses were allocated 1, 2, 3, or 4 points based on the answer selection, and an individual's cumulative anxiety score was derived from the total points accrued from the 30 test items.

## Results

All collected data and analysis scripts can be accessed on the Open Science Framework (https://osf.io/p64yf/). Table 2 offers a descriptive statistics overview of the various variables employed in our analyses, whereas Table 3 provides correlation coefficients among all the variables. Crucially, no correlations were detected among the three cognitive variables: language proficiency, working memory, and anxiety. In fact, we employed a stepwise VIF (variance inflation factor) selection procedure wrapped around the vif function in the fmsb R package (for details on the stepwise VIF selection procedure, refer to https://www.r-bloggers.com/collinearity-and-stepwise-vif-selection/) to discern any potential collinearity among the three cognitive variables. Not a single VIF surpassed 1.12, indicating no substantial evidence of serious collinearity among these cognitive predictors. Consequently, they were all incorporated into subsequent regression analyses.

We applied multiple regression in our investigation of total errors and errors across various linguistic (conceptual, lexical, syntactic, and phonological) levels as a function of an interpreter's language proficiency, working memory, and anxiety. All three continuous predictors were scaled, meaning they were transformed into z-scores. As demonstrated in Fig 1 and Table 4, total errors decreased as a function of language proficiency and working memory but increased as a function of anxiety.

At different linguistic levels, conceptual errors emerged as the most common. It should be noted that in the context of interpreting, interpreters do not independently conceptualize; they simply convey messages from the source language. Hence, conceptual errors originated from misunderstandings of the source speech in English, the interpreters' second language (L2). Conversely, phonological errors were the least frequent, given that the target language was interpreters' L1. As revealed in Fig 1 and Table 4, conceptual errors decreased as a function of both language proficiency and working memory but did not change as a function of anxiety. Lexical errors did not change as a function of language proficiency or working memory but

**Table 2. Descriptive results of the cognitive factors, interpreting rating and error rates (number of errors out of 1000 characters in the interpreting output).**

|  | Range (full score) | Mean | SD |
|---|---|---|---|
| Language proficiency | 33–44 (56) | 37.9 | 3.0 |
| Working memory | 50–91 (100) | 73.1 | 9.3 |
| Anxiety | 33–93 (120) | 65.8 | 12.2 |
| Total error rate | 11.3–46.3 (NA) | 25.3 | 9.6 |
| Conceptual error rate | 2.2–21.2 (NA) | 9.4 | 5.2 |
| Lexical error rate | 1.5–16.6 (NA) | 7.2 | 3.4 |
| Syntactic error rate | 1.0–11.6 (NA) | 5.5 | 2.5 |
| Phonological error rate | 0–2.7 (NA) | 0.4 | 0.6 |

**Table 3. Correlation coefficients among the measured variables.**

|  | WM | Anxiety | Total | Conceptual | Lexical | Syntactic | Phonological |
|---|---|---|---|---|---|---|---|
| Proficiency | -0.03 | -0.27 | -0.47*** | -0.48*** | -0.25 | -0.36** | 0.03 |
| WM |  | -0.19 | -0.19 | -0.34* | 0.02 | -0.12 | 0.16 |
| Anxiety |  |  | 0.42** | 0.29* | 0.4** | 0.42** | -0.21 |
| Total |  |  |  | 0.84*** | 0.77*** | 0.71*** | 0.06 |
| Conceptual |  |  |  |  | 0.39** | 0.43** | -0.13 |
| Lexical |  |  |  |  |  | 0.47*** | 0.13 |
| Syntactic |  |  |  |  |  |  | -0.02 |

Note: *, ** and *** respectively refer to *p*-values smaller than .05, .01, and .001. *WM* = working memory, *Total*, *conceptual*, *lexical*, *syntactic*, and *phonological* respectively refer to the error rate at each linguistic level.

increased as a function of anxiety. Syntactic errors decreased as a function of language proficiency and increased as a function of anxiety, though it did not change according to working memory. Finally, phonological errors did not vary as a function of any of the cognitive factors.

Finally, we probed the relative significance of the three cognitive predictors on error frequency. In this endeavor, we utilized the lmg metric (available in the relaimpo package in R), which enabled us to estimate the relative importance of predictors in explaining observed data [67, 68]. Fig 2 presents the relative importance of the three cognitive predictors in each model. Of the three predictors, language proficiency played the most crucial role in the occurrence of total errors and syntactic errors, with some impact on lexical and syntactic errors and a very minimal influence on phonological errors. Working memory played a more restrained role, exerting some influence on the occurrence of conceptual and phonological errors and very minimal impact on total, lexical, and syntactic errors. Anxiety held the most significant influence in the occurrence of lexical, syntactic, and phonological errors, with a relatively less significant role in the occurrence of total and conceptual errors.

## Discussion

In this study, we investigated how individual variations among student interpreters in language proficiency (in the L2 source language), working memory, and anxiety influence their speech errors during English-Chinese consecutive interpreting. Broadly, total errors in target language output diminished as a function of the interpreter's proficiency in English (L2 source language), remained constant in relation to working memory, and surged as a function of the interpreter's general anxiety (refer to Table 5 for a summary). Dissected at various linguistic levels, conceptual errors decreased as a function of language proficiency and working memory but persisted unaltered by anxiety. Conversely, lexical errors were unaffected by language proficiency or working memory but escalated with increasing anxiety. Syntactic errors decreased as a function of language proficiency and increased as a function of anxiety, exhibiting no fluctuation with working memory. Ultimately, phonological errors remained stable irrespective of the three cognitive capacities. It's crucial to note, though, that the absence of these impacts was likely due to the limited quantity of phonological errors made when translating into the interpreter's L1 (Chinese).

### Effects of language proficiency

Language proficiency in this context denotes the competency level in our student interpreters' L2 (English), which served as the source language in the interpreting task. Consequently, in our investigation, this factor primarily signifies the capacity for source language

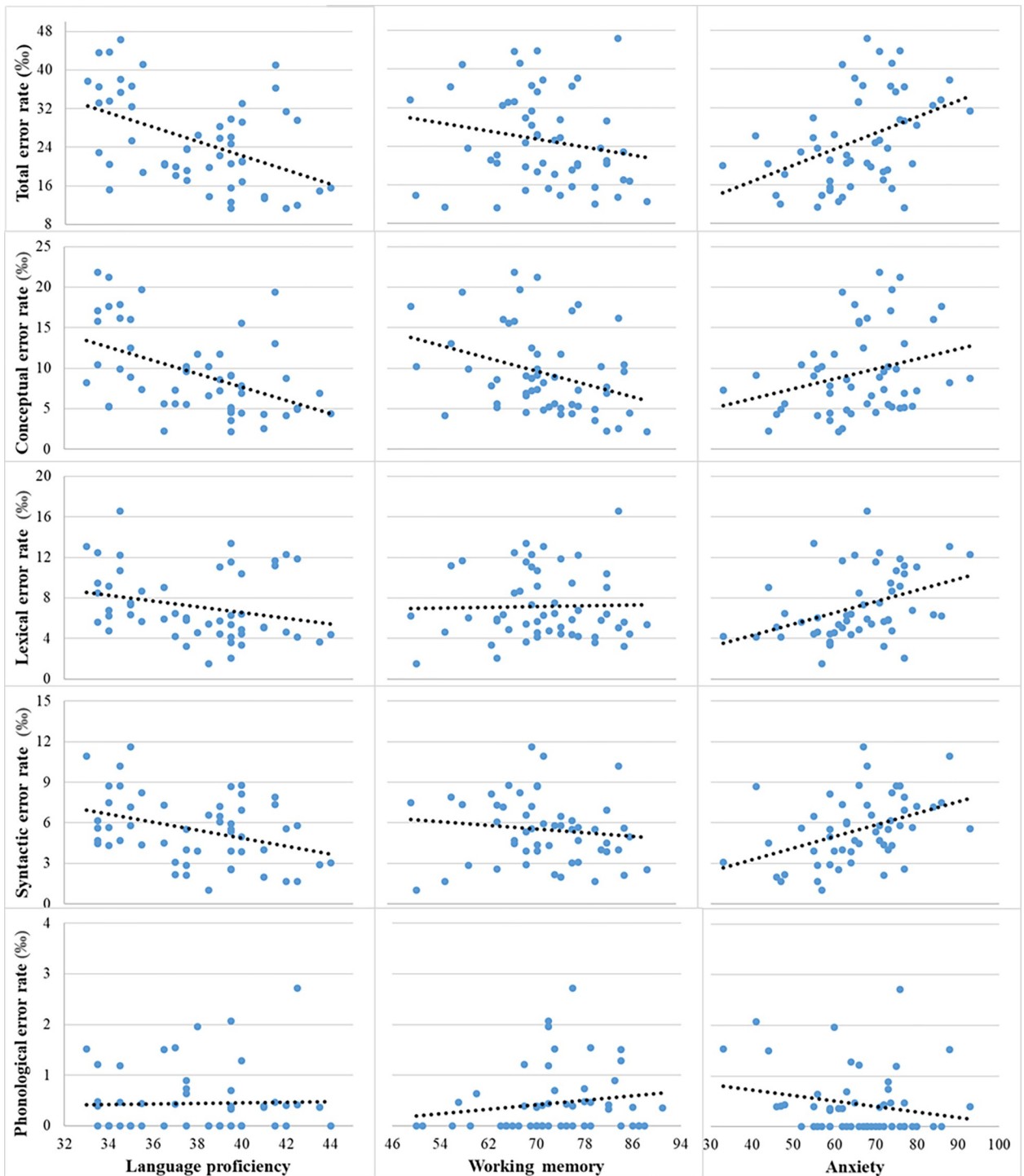

**Fig 1. Number of errors per 1000 characters at different linguistic levels as a function of language proficiency, working memory and anxiety.**

comprehension in interpreting. Established research attests that L2 comprehension improves as a function of proficiency [refer to 69, for a review]. Inherently, conceptual errors in target language production stem from comprehension failures of the source language (English in our case); hence, more proficient interpreters in the L2 source language generate fewer conceptual

**Table 4. Errors at different linguistic levels as a function of language proficiency (LP), working memory (WM) and anxiety.** *P*-values are marked bold for significant effects.

| | Total error | | | Conceptual error | | | Lexical error | | | Syntactic error | | | Phonological error | | |
|---|---|---|---|---|---|---|---|---|---|---|---|---|---|---|---|
| | *β* | *t* | *p* | *β* | *t* | *p* | *β* | *t* | *p* | *β* | *t* | *p* | *β* | *t* | *p* |
| LP | -1.25 | -3.24 | **.002** | -0.79 | -3.90 | **< .001** | -0.17 | -1.24 | .266 | -0.22 | -2.09 | **.042** | -0.004 | -0.13 | .894 |
| WM | -0.16 | -1.28 | .206 | -0.19 | -2.89 | **.006** | 0.03 | 0.69 | .494 | -0.02 | -0.54 | .590 | 0.009 | 0.89 | .376 |
| Anxiety | 0.23 | 2.34 | **.023** | 0.04 | 0.82 | .419 | 0.10 | 2.77 | **.008** | 0.07 | 2.55 | **.014** | -0.010 | -1.27 | .210 |

errors in their output. Language proficiency also notably influenced syntactic errors, with more proficient interpreters generating fewer syntactic errors. It's plausible that interpreters with greater proficiency in the L2 source language possess a superior understanding of the source language meaning, hence less prone to false starts or sentence reorganisation in their output, leading to fewer syntactic errors.

## Effects of working memory

A wealth of evidence indicates the significant role resources in language comprehension (e.g., [70, 71]) and language production [72, 73], including in L2 processing [see 74, for a review]. Our observations are in harmony with these findings. Specifically, we found that working memory impacts the frequency of conceptual errors in interpreting. Interpreters with a larger working memory span committed fewer conceptual errors, implying a greater aptitude at comprehending the L2 source language.

Our findings regarding the effects of working memory align with previous findings about its influence on interpreting performance [22, 34, 35, 46, 63, 75]. Moreover, our data suggests that, akin to language proficiency, working memory in comprehension restricts the conceptual understanding of the source language. An intriguing question arises as to why working

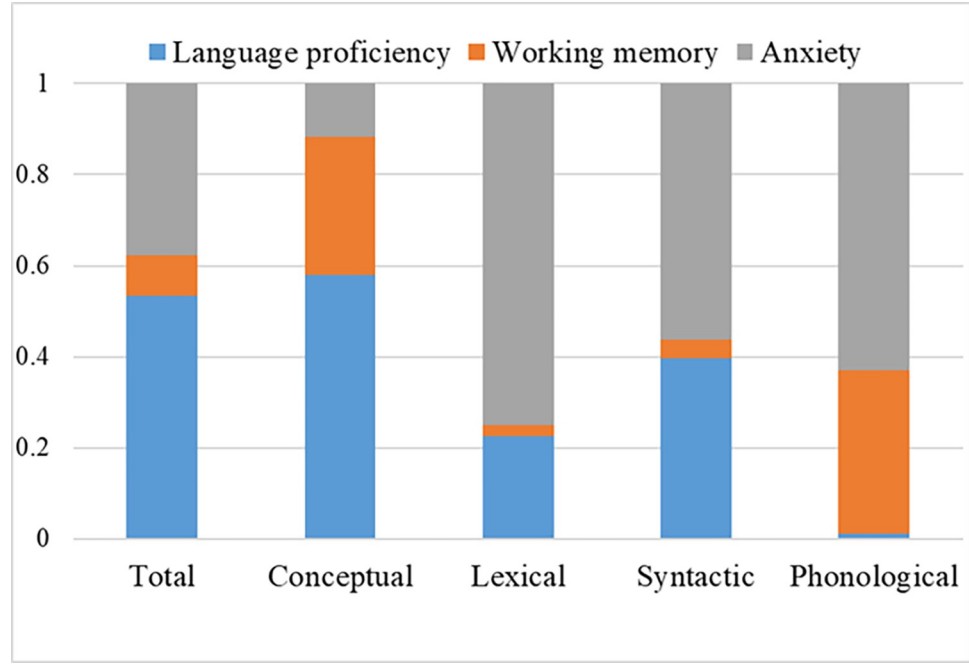

**Fig 2. Relative importance of the three cognitive factors in error occurrence at different lingusitic levels.**

**Table 5. Summary of the effects of cognitive factors on speech monitoring.** ↑ indicates the error rates increased as function of a predictor while ↓ indicates error rates decreased as a function of a predictor.

| Error type | Language proficiency | Working memory | Anxiety |
|---|---|---|---|
| Total | ↓ | | ↑ |
| Conceptual | ↓ | ↓ | |
| Lexical | | | ↑ |
| Syntactic | ↓ | | ↑ |
| Phonological | | | |

memory, unlike language proficiency, only impacts conceptual understanding of the source language without affecting error occurrence at other linguistic levels. We propose two potential explanations, both of which require empirical verification. First, lexical access, syntactic encoding, and phonological encoding might be relatively automatic processes and, thus, scarcely affected by individual differences in working memory in interpreters. Second, the absence of working memory effects on lexical and syntactic errors might result from the note-taking practice commonly adopted during interpreting (a practice our participants engaged in during the experiment). The use of notes as an external memory source bolsters interpreting accuracy [76], substantially mitigating potential working memory effects on lexical and syntactic errors.

## Effects of anxiety

Anxiety can impact attention and cognitive functions, such as shifting, inhibition, and updating [77]. Specifically, an anxious individual's attention tends to be stimulus-driven rather than goal-driven. Within interpreting, this suggests an anxious interpreter is more likely to be consumed by audience reactions than the interpreting task at hand (i.e., language comprehension and production). Furthermore, anxiety can impair cognitive functions by reducing one's capacity to shift between tasks (e.g., from lexical processing to syntactic processing during interpreting), and to suppress irrelevant and interfering information (e.g., source language comprehension might interfere with target language production via cross-language structural or phonological priming; e.g., [25, 30]).

Given these functions, anxiety appears to influence various aspects of speech errors in interpreting, a conclusion that echoes previous research on the role of anxiety in student interpreters' competence [40, 50–55, 65]. Particularly, interpreters with higher anxiety levels tend to commit more lexical and syntactic errors. Two possible reasons can explain this relationship. First, as anxiety diminishes goal-directed attention, more anxious interpreters may not allocate sufficient attention to lexical and syntactic processing during target language production, leading to more errors. Second, anxiety also weakens the effectiveness of inhibition; thus, more anxious interpreters are less able to inhibit erroneous translation equivalents and syntactic structures activated during source language comprehension (e.g., [20, 21]), thereby increasing the likelihood of lexical and syntactic errors.

## Pedagogical implications

Our study underscores the traditional understanding that language proficiency and working memory are vital facets of interpreting training. More importantly, it illustrates that, particularly among student interpreters, anxiety has a more significant role in interpreting than previously perceived. According to the attentional control theory of anxiety [77], anxiety leads to more stimulus-driven attention (attention to the irrelevant environment) and less goal-

directed attention (attention for completing the interpreting). It also impairs individuals' executive functions including shifting, inhibition, and updating. Therefore, reducing students' anxiety levels (especially in public) should be a crucial component of the interpreting training curriculum. To achieve this, teachers could evaluate each student interpreter's anxiety level (e.g., [65]). More anxious students should be granted additional opportunities to speak in public.

Our study also highlights that student interpreters frequently produce speech errors, notably errors stemming from mistakes in source language comprehension (conceptual errors), poor lexical choices (lexical errors), and improper organization of target language sentences (syntactic errors). Thus, student interpreters should be made aware of these error types. Metacognitive self-regulation training is highly recommended; there is evidence that such training can improve translation (e.g., [78]). It's plausible that similar metacognitive self-regulation training in interpreting could help reduce speech errors.

## Conclusion

We demonstrated that, in consecutive interpreting from English (L2) to Chinese (L1), student interpreters committed the most speech errors pertaining to source language comprehension (i.e., conceptual errors), followed by syntactic and lexical errors, with the least speech errors in phonology. We further demonstrated that interpreters committed fewer conceptual and syntactic errors if they had superior proficiency in the L2 English, committed fewer conceptual errors if they had larger working memory spans, and committed fewer lexical and syntactic errors if they were less anxious. Our results underscore the importance of anxiety reduction in interpreting training.

## Author Contributions

**Conceptualization:** Nan Zhao, Zhenguang G. Cai, Yanping Dong.

**Data curation:** Nan Zhao, Zhenguang G. Cai.

**Formal analysis:** Nan Zhao, Zhenguang G. Cai.

**Funding acquisition:** Zhenguang G. Cai, Yanping Dong.

**Investigation:** Nan Zhao, Zhenguang G. Cai.

**Methodology:** Nan Zhao, Zhenguang G. Cai.

**Project administration:** Nan Zhao.

**Visualization:** Nan Zhao, Zhenguang G. Cai.

**Writing – original draft:** Nan Zhao, Zhenguang G. Cai.

**Writing – review & editing:** Nan Zhao, Zhenguang G. Cai, Yanping Dong.

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
