## [Decision Letter · Decision Letter 0]

22 Mar 2022

PONE-D-21-19586Speech errors in consecutive interpreting: Effects of language proficiency, working memory, and anxietyPLOS ONE

Dear Dr. Zhao,

Thank you for submitting your manuscript to PLOS ONE. After careful consideration, we feel that it has merit but does not fully meet PLOS ONE’s publication criteria as it currently stands. Therefore, we invite you to submit a revised version of the manuscript that addresses the points raised during the review process.

The reviewers are in agreement that your study is a novel examination of important factors and is relevant to the field. They raise various points for clarification adn improvement. Of particular importance with respectto PLOS ONE’s publication criteria is clear and complete reporting of methods and results, and making clear how your conclusions are supported by the data.

We look forward to receiving your revised manuscript.

Kind regards,

Daniel Mirman

Academic Editor

PLOS ONE

Journal Requirements:

2. Thank you for including your ethics statement:  "The research was ethically approved by an institutional board.".   

a.) Please amend your current ethics statement to include the full name of the ethics committee/institutional review board(s) that approved your specific study. 

b.) Please provide additional details regarding participant consent. In the ethics statement in the Methods and online submission information, please ensure that you have specified (1) whether consent was informed and (2) what type you obtained (for instance, written or verbal, and if verbal, how it was documented and witnessed). If your study included minors, state whether you obtained consent from parents or guardians. If the need for consent was waived by the ethics committee, please include this information.

For additional information about PLOS ONE ethical requirements for human subjects research, please refer to " ext-link-type="uri" xlink:type="simple">http://journals.plos.org/plosone/s/submission-guidelines#loc-human-subjects-research."

3. Please change "female” or "male" to "woman” or "man" as appropriate, when used as a noun (see for instance https://apastyle.apa.org/style-grammar-guidelines/bias-free-language/gender).

Z.G.C was supported by a General Research Fund (14600220) from the University Grants Committee of Hong Kong SAR.

Reviewers' comments:

Reviewer's Responses to Questions

**Comments to the Author**

1. Is the manuscript technically sound, and do the data support the conclusions?

Reviewer #1: Yes

Reviewer #2: Partly

Reviewer #3: Yes

2. Has the statistical analysis been performed appropriately and rigorously? 

Reviewer #1: Yes

Reviewer #2: Yes

Reviewer #3: Yes

3. Have the authors made all data underlying the findings in their manuscript fully available?

Reviewer #1: Yes

Reviewer #2: No

Reviewer #3: Yes

4. Is the manuscript presented in an intelligible fashion and written in standard English?

Reviewer #1: Yes

Reviewer #2: No

Reviewer #3: No

5. Review Comments to the Author

Reviewer #1: This article is well-written. I just have some concerns on the rationale of using the the paradigm as WM test. It seems this test is highly dependent on language learners' proficiency level. I am quite surprised of not seeing the correlation results among the examined variables but directly the relative significance of each variable. The discussion section did not consider conceptual errors, syntactic errors, and phonological errors.

Reviewer #2: The study is novel and generally of interest for the field of interpreting studies. The empirical part of the study is quite neat, and the analysis is reliable and clearly described.

My major concern is mainly due to the cumulative effect of quite a number of minor concerns. After reading the paper, it was not 100% clear to me what the theoretical motivation of the study was, or what its implications were. Certainly, the discussion section and the motivation section did not tie in well together, and at times they did not tie in clearly with the results. I believe that before publication, the authors need to clearly motivate the study, define the terms used in the study, and then relate the discussion section back to the motivation section of the study.

There are also a number of places where flimsy claims are made that are either not logical within the paper, or else are not backed up with any citations.

Data availability

Appendices 1 and 3 are mentioned but were not in the version I received. Also, Appendix 2 is not mentioned so the numbering appears to be incorrect.

Reference missing for Spielberger (1983). I did not check all the references – I tried to check this one since I could not find the anxiety scale on the OSF page.

Anxiety scale is not on the OSF page.

Inaccessibility of parts of the paper for non-Chinese readers.

Some information was not accessible for me as a non-Chinese speaker. There are various examples provided throughout the text in Chinese only. An English translation (word-by-word if necessary) should be provided for all of these examples.

I was also unable to understand any of the examples of types of error in Table 1

Use of terms

Cognitive resources vs. cognitive processes are not defined, working memory is not defined, anxiety is not defined, language proficiency is not clearly related to proficiency in the second language (which is in fact what this paper investigates).

The aims of the study

The aims of the study do not always match either the empirical part of the study or the literature review. For instance:

p3: the claim is made that the research aims to shed light on language production and speech monitoring mechanisms. However, I do not find reference to this again and it is not clear how this could be done via the study described in the paper.

In general: none of the aims of the study would involve assessing interpreting performance. This is not mentioned in either the title, abstract, motivation or conclusions of the study, and so it is surprising when on p11, we learn that performance was assessed.

There is little motivation for why the authors hypothesized that errors at different levels would relate differently to any of the mentioned “cognitive factors”.

Literature review

P6: Do the sources cited (Galantucci, Fowler Turvey; Liberman Mattingly) mention interpreters specifically or do the authors of the paper point out that this thus extends to interpreters listening in the source language? This should be made clear.

P3 and p4 – Logic: page 3 says that interpreters and speakers go through the same conversion stages (e.g., conceptualizing a message, the meaning they wish to convey) whereas p5 says that this is a difference between interpreting and speaking. This should be clarified or rephrased on page 3.

P7: Title of section “interpreting and cognitive resources” does not match content of section, which discusses the role of language proficiency, working memory, anxiety in interpreting and then the influence of these factors on speech errors.

NB: The sources cited in this section generally refer to simultaneous interpreting while the study is on consecutive interpreting (e.g., Gile 2009, Gerver 1974). Would it be possible to include more sources on consecutive interpreting?

P7 – citations required for claims “all of these cognitive factors have been examined in interpreting research” and “due to their close relevance to interpreting performance”.

P8 – citation for the claim “such anxiety can become one of the major obstacles in the early stages of interpreting training.”

P8/9: There is a bit of a mix in the last paragraph of page 8 and then moving onto page 9 between the impact of different cognitive factors on speech production in interpreting and in monolingual production. In general, there is not much evidence presented here for what is ultimately the main focus of the study.

P9: Sentence unclear “better in terms of interpreting disfluency” – presumably the authors mean that interpreters with higher WM produce more fluent interpretations?

P9: first language appears to be equivalent to source language and non-native language, and second language is equivalent to target language or native language, which does not correspond to usual use of L1, L2 etc. Then on p10, first language once again means native language (as I think it does in the rest of the paper too).

P10: I do not follow the logic here. The authors previously introduced production errors in interpreting as being different from those in “normal” production because they are directly related to source language comprehension (and this is also why language proficiency in the source language is considered). Now the authors claim that consecutive interpreting is more akin to spontaneous language production and therefore more revealing of different processing stages in target language production. But this goes against the logic put forward previously. I also do not follow why speech errors in consecutive interpreting would be more revealing of the influences of different cognitive factors than speech errors in simultaneous interpreting.

P10 – Logic: “The research will have important pedagogical implications” because participants are student interpreters – a study that uses students as participants does not automatically have pedagogical implications.

Methods

P11 – It would be helpful to state that SCIC is the European Commission’s interpretation service. It is unclear what the scale described is. The source of the original scale (Geneva University/SCIC) should be cited. If the scale only includes Accuracy/Completeness, Target Language Expression and Proper Delivery and Interpreting Strategies and Manner, then it appears to be underspecified. How were the percentages decided upon?

P12: Under Transcription and error coding – it is unclear how errors would arrive at a cognitive process. What does this mean?

P13: English language test. Had participants already completed this test or did they complete this test as part of their studies or as part of this study in particular?

The text should make it clear that the anxiety scale used is a combination of Zhang and Schwarzer’s self-efficacy test and Spielberger’s trait anxiety test (I understand this to be the case).

Results/Discussion

The analysis should state what percentage of errors were conceptual, what percentage were syntactic etc.

P16. Presumably it should read that total error rate increased as a function of anxiety level?

P18: Conceptual errors arose due to misunderstanding: given that the authors previously show that conceptual errors also occur during “normal” speaking, surely it is not now possible to conclude that all conceptual errors in consecutive interpreting are the result of misunderstanding (see also my comment re. p10 and the authors motivation for their decision to study consecutive interpreting).

P18: As previously mentioned “language proficiency” needs to be further specified. Here it is particularly relevant, because the authors now include syntactic, lexical and phonological errors and how they relate to language proficiency – but are assessing these errors in the Chinese output after having tested language proficiency in English and not in Chinese

Discussion

Discussion p23. “More errors in understanding the conveyed message lead to poorer interpreting performance” – previously in the results, evidence of a negative correlation between error rate and performance was established, so it seems odd to conclude here that there was a cause/effect.

Also, here, the authors conflate conceptual production errors and errors in understanding. I do not believe that comprehension can be directly measured through production in this way.

Discussion p24, first paragraph

It seems unusual to add in new citations in the discussion section, rather than referring back to the motivation section of the study. Perhaps this part of the discussion would have been better placed in the motivation section.

It is unclear how the results suggest that working memory affected interpreting performance via conceptual understanding, as language proficiency did. The results presented provide an overall rating for interpreting performance based on a scale comprising conceptual, stylistic and presentation elements but do not break down the interpreting performance according to the scale.

P24: Discussion on why working memory and language proficiency only affect conceptual understanding – the two possible accounts here only mention working memory but do not offer any explanation re. language proficiency.

It is unclear for me why syntactic and phonological information would be “interfering” with understanding of the source language?

P24: why are we now discussing error repairs when the authors previously state that error repairs are not addressed.

P26: Conclusion? It would be helpful to conclude summing up very briefly what the study has shown.

Style and grammar in the paper

The reader can generally always understand the meaning of the text, but there are a number of errors in English that need to be addressed. Below I have noted the errors over the first six pages but note that there are errors throughout.

Abstract, line 12, word missing: Lexical ___ increased as a function of the interpreter’s tendence of anxiety.

First sentence unclear – “interpreter relays a message in the source in the target language”.

Line 5, word missing: “the current paper __ whether and how”

P3: Under language production. “In speaking (and also interpreting) the speaker converts a pre-verbal message...” This should perhaps be clarified as interpreting involves both speaking and listening, so perhaps should say e.g., during the production phase of interpreting

P4, line 7: “speakers may say “the star” to refer to a small square without realizing there is a bigger star” – does not appear to make sense, please check.

P4, line 15-16 (e.g., children aged 5 than 7) should read e.g., for children aged five compared to children aged seven.

P5, under Speech errors in interpreting – “Interpreting largely goes through the same cognitive routines as in regular language production” – rephrase (interpreting is a process and cannot go through a cognitive routine).

P5, turn of phrase “Unlike in regular language production (change to unlike regular language production).

P6: Example given seems odd “the reading of the electricity” is more usually called the “meter reading” in English.

Reviewer #3: I have included an attachment with comments to the authors. I believe that this manuscript presents a novel study examining the effects of three important factors (language proficiency, anxiety, and working memory) on errors in consecutive interpreting. In the attached document, I have included my suggestions on needed revisions as well as some questions to the authors. If these are properly addressed, then I would recommend the manuscript for publication in your journal.

6. PLOS authors have the option to publish the peer review history of their article (what does this mean?). If published, this will include your full peer review and any attached files.

Reviewer #1: **Yes: **Mark Feng Teng

Reviewer #2: No

Reviewer #3: **Yes: **Ena Hodzik

---

## [Author Response · Author response to Decision Letter 0]

26 Jun 2023

See the file "PONE Reviews R1 responses".

---

## [Decision Letter · Decision Letter 1]

14 Jul 2023

PONE-D-21-19586R1Speech errors in consecutive interpreting: Effects of language proficiency, working memory, and anxietyPLOS ONE

Dear Dr. Zhao,

Thank you for submitting your manuscript to PLOS ONE. After careful consideration, we feel that it just needs a bit more explanation and clarity about the relationship between these results and the ones reported in Zhao (2022) - Reviewer #2 has articulated the specific points quite clearly.

We look forward to receiving your revised manuscript.

Kind regards,

Daniel Mirman

Academic Editor

PLOS ONE

Journal Requirements:

Reviewers' comments:

Reviewer's Responses to Questions

**Comments to the Author**

1. If the authors have adequately addressed your comments raised in a previous round of review and you feel that this manuscript is now acceptable for publication, you may indicate that here to bypass the “Comments to the Author” section, enter your conflict of interest statement in the “Confidential to Editor” section, and submit your "Accept" recommendation.

Reviewer #2: (No Response)

Reviewer #3: All comments have been addressed

2. Is the manuscript technically sound, and do the data support the conclusions?

Reviewer #2: Yes

Reviewer #3: Yes

3. Has the statistical analysis been performed appropriately and rigorously? 

Reviewer #2: Yes

Reviewer #3: Yes

4. Have the authors made all data underlying the findings in their manuscript fully available?

Reviewer #2: Yes

Reviewer #3: Yes

5. Is the manuscript presented in an intelligible fashion and written in standard English?

Reviewer #2: Yes

Reviewer #3: Yes

6. Review Comments to the Author

Reviewer #2: Review: Speech errors in consecutive interpreting: Effects of language proficiency, working memory, and anxiety

Thank you to the authors for having addressed all of the comments I made in my first review. Overall, the manuscript is much improved. At this stage my only concern is the lack of transparency in the revised article about its link to Zhao, 2022.

As the authors acknowledge in the Response to Reviewers, Zhao 2022 and this new manuscript deal with different data from the same study.

Therefore, I believe that in the "Methods" section of this article, the authors should state that the data come from the same study as Zhao, 2022.

Additionally, Zhao (2022) is the only support for the claim on p7 that "these cognitive functions have been extensively examined in interpreting research due to their relevance to interpreting performance". I appreciate that more detail is provided about some other studies in the following paragraphs, but I still believe several citations should be included directly after this claim if the claim "extensively studied" is made.

The authors have also added a paragraph on p9 about Zhao, 2022. If the authors wish to include this paragraph, I believe it should again be clear here that Zhao, 2022 reports different results from the same study. The wording currently used ("recent studies") suggests that various different studies have been carried out in this area. Similarly, citing Zhao, 2022 on p10 as evidence that more anxious interpreters are more disfluent and that this can lead to more speech errors suggests again that the results of Zhao (2022) are independent from those of the present study.

Once there is more transparency about the link between Zhao 2022 and this new paper, I do not believe any other changes need to be made.

Reviewer #3: (No Response)

7. PLOS authors have the option to publish the peer review history of their article (what does this mean?). If published, this will include your full peer review and any attached files.

Reviewer #2: No

Reviewer #3: **Yes: **Ena Hodzik

---

## [Editor Report · Decision Letter 2]

27 Sep 2023

Speech errors in consecutive interpreting: Effects of language proficiency, working memory, and anxiety

PONE-D-21-19586R2

Dear Dr. Zhao,

We’re pleased to inform you that your manuscript has been judged scientifically suitable for publication and will be formally accepted for publication once it meets all outstanding technical requirements.

Within one week, you’ll receive an e-mail detailing the required amendments. When these have been addressed, you’ll receive a formal acceptance letter and your manuscript will be scheduled for publication. It should be possible to fix the funding statement during this stage as well. Also, there is a small typo near the bottom of p. 3: "semantic inference" should be "semantic interference".

Kind regards,

Daniel Mirman

Academic Editor

PLOS ONE
---

## [Editor Report · Acceptance letter]

9 Oct 2023

PONE-D-21-19586R2 

Speech errors in consecutive interpreting: Effects of language proficiency, working memory, and anxiety 

Dear Dr. Zhao:

I'm pleased to inform you that your manuscript has been deemed suitable for publication in PLOS ONE. Congratulations! Your manuscript is now with our production department. 

Kind regards, 

on behalf of

Dr. Daniel Mirman 

Academic Editor

PLOS ONE